# Knowledge, attitudes and practices of healthcare workers towards noma in Zambezia, Mozambique

Marta Ribes[1,2]*, Fizaa Halani[2], Abdala Atumane[3], Milagre Andurage[4], Eldo Elobolobo[5,6], Gemma Moncunill[1,2,7], Romina Rios-Blanco[2,8], Tairo Sumine[9], Luis Transval[9], Fernando Padama[3☉], Carlos Chaccour[1,7,10☉]

1 ISGlobal, Barcelona, Spain, 2 Facultat de Medicina i Ciències de la Salut, Universitat de Barcelona (UB), Barcelona, Spain, 3 Serviço Provincial de Saúde da Zambezia, Zambezia, Mozambique, 4 Hospital Central de Quelimane, Quelimane, Mozambique, 5 Associação Silver Lining, Quelimane, Mozambique, 6 Databrew, Chimoio, Mozambique, 7 CIBER de Enfermedades Infecciosas, Madrid, Spain, 8 Instituto de Medicina Tropical Alexander von Humboldt, Universidad Peruana Cayetano Heredia, San Martin de Porres, Perú, 9 Hospital Geral de Quelimane, Quelimane, Mozambique, 10 Navarra Center for International development, Universidad de Navarra, Pamplona, Spain

☉ These authors contributed equally to this work.
* marta.ribes@isglobal.org

⛓ OPEN ACCESS

## Abstract

### Background

Noma is a neglected tropical disease primarily affecting children living in poverty. Despite being preventable and treatable with readily accessible medicines, an estimated 90% of patients die due to lack of access to prompt and appropriate care.

### Methodology and principal findings

Primary, secondary and quaternary health facilities were visited on a convenience-sampling basis in Zambezia Province, central Mozambique. Health professionals were invited to participate on a quota-sampling basis, and were administered a questionnaire including open and close-ended questions assessing their oral health practices, theoretical knowledge on noma and attitudes towards receiving a noma training.
A total of 41 health professionals from 23 different health facilities participated in the study. Of these, 59% were aware of noma, and 26.8% reported having personally attended to an acute noma patient. However, their knowledge of noma's clinical characteristics and management was poor, especially in its early stages. Only 12% correctly diagnosed noma at stage 1, and 5% at stage 2. University-level professionals had a significantly better understanding of the disease than nurses and technicians. All participants were keen on receiving specific noma training.

### Conclusions

Noma management competencies in Zambezia are extremely low, particularly among nurses and medical technicians, who serve as the first point of care for noma patients.

**Data availability statement:** Anonymized, individual participant data is available at https://github.com/marta-ribes/Noma-Echoes-KAP-questionnaire.git.

**Funding:** This study was funded by ISGlobal and the University of Navarra through a crowd-funding campaign. MR received salary from an AGAUR-FI_B 01022 PhD fellowship from the Catalan Government and the European Social Fund. GM received salary from RYC2020-029886-I/AEI/10.13039/501100011033, co-funded by the European Social Fund (ESF). We acknowledge support to ISGlobal from the grant CEX2023-0001290-S funded by MCIN/AEI/ 10.13039/501100011033, and support from the Generalitat de Catalunya through the CERCA Program. The funders had no role in study design, data collection and analysis, decision to publish, or preparation of the manuscript.

**Competing interests:** The authors have declared that no competing interests exist.

There is an urgent need to implement comprehensive training programs across all levels of Mozambican healthcare providers, to prevent further avoidable deaths and reduce the severe outcomes associated with delayed treatment.

## Author summary

Noma is a disease that mostly affects children living in poverty. It starts as a form of gingivitis that, without treatment, rapidly evolves into the death of the surrounding tissues, including the skin and bones. Survivors often face severe facial disfigurement, which affects their well-being, from eating or speaking, to socialising. However, noma is preventable. If healthcare workers could recognize it early, a simple course of antibiotics, wound care and proper nutrition—things that are widely available—could stop the disease before it causes such damage.

For this reason, we wanted to know the level of awareness of noma among health practitioners in Zambezia, Mozambique. We found that while 59% of them had heard about noma, and 26.8% had treated a patient themselves, very few could diagnose it or treat it early enough to prevent its irreversible consequences, or were aware of its fatality and rapid progression. This is similar to findings in Zambia and Burkina Faso, where noma has also been neglected.

This indicates an urgent need to raise awareness and provide training for healthcare workers. The issue stems not from a lack of access to treatment, but rather from a lack of access to a timely diagnosis.

## Introduction

Noma (*cancrum oris*), is a rapidly progressing orofacial necrosis, that mostly affects children living in poverty and less often immunocompromised adults. Although readily and easily treatable at its early stages with antibiotics, nutritional support and wound debridement, most cases go undiagnosed and therefore untreated, leading to an estimated mortality of 90% within just two weeks [1,2]. Most survivors of the acute phase live with lifelong facial disfigurement that leads to tremendous social and functional consequences, including stigma, isolation and difficulties eating, speaking or seeing [1–3].

Noma's aetiology is still unknown, although thought to be the result of an imbalance in the oral microbiota[4,5]. It is not contagious, nor recurrent, and the most commonly reported risk factors are malnourishment, poor oral hygiene, and having recently suffered from another infection such as measles or malaria within the previous three months[1]. Its current epidemiology is not completely understood as there is a striking lack of reporting [6]. However, it is thought to occur wherever there is poverty, as corroborated by Srour et al. and our team, who met noma survivors in Laos [7] and Mozambique, respectively, countries which had not previously reported any cases in the scientific literature.

Noma's progression has been categorised by the World Health Organization (WHO) in five stages [2], preceded by the warning signal or stage 0, which corresponds to a simple gingivitis. Stage 1 describes an acute necrotising gingivitis, characterised by fetid breath, ulceration of one or more interdental papillae with pain, spontaneous bleeding of gums and excessive salivation. At this stage, administration of a combination of oral amoxicillin and metronidazole,

accompanied by analgesics, wound care and debridement, and nutritional support, stops the progression of the disease. In absence of appropriate treatment, stage 2 rapidly follows, characterised by an oedema on the exterior side of an inner ulcer which extends rapidly, accompanied by intense pain and high fever. Treatment consists of intravenous antibiotics (amoxicillin clavulanic acid, penicillin or ampicillin, with gentamicin and metronidazole), hydration and nutrition, and debridement. Up to this stage the disease is reversible; however, in the absence of treatment, in a matter of hours to days, it evolves into stage 3 the gangrenous stage. During this stage the oedema progresses to necrosis, affecting soft and hard tissue, which eventually sloughs off, leaving a perforation in the affected area. At this point, the management of the patient consists of the previously mentioned, with the priority of stabilizing the patient. Death usually occurs at this stage because of sepsis, bronchoaspiration or malnutrition. Stage 4 describes the scarring process after the necrosis has been arrested, and stage 5 refers to the chronic sequelae, only reversible with specialised surgery [8].

Noma was only officially recognized as a Neglected Tropical Disease (NTD) in December 2023 [9]. Indeed, lack of awareness has surrounded noma from academia to health systems [10]. Noma is absent from most curricula and public health policies. As a result, health practitioners' knowledge is far from optimal. To our knowledge, four research studies have evaluated the level of knowledge and practice competence of health professionals on noma. In 2009 in Serenje District, Zambia, among a cohort of 35 healthcare workers with pre-university level, Ahlgren et al. found that 54% had heard about the disease and all of them had a suboptimal or very low level on overall practice competence and two-thirds, a very low level of theoretical knowledge[11]. In 2009 in Nouna District, Burkina Faso, Brattström-Stolt et al. administered a structured questionnaire to 76 nurses and nurse assistants and found that 91% of them had heard about noma, but 70% of them had suboptimal or very low practice competences, while half of participants had good or optimal theoretical knowledge. A subgroup of nurses who received a two-day training had a better level of knowledge than their peers [12]. In 2019 Bala et al. evaluated 156 professionals from the Usmanu Danfodiyo University Teaching Hospital, a tertiary institution in Sokoto state, Nigeria, located just five kilometers away from Noma Children Hospital [13], and found an awareness rate of 79% [14]. In 2022, the same team interviewed 251 primary health workers and found that 83.7% were aware of noma, and among them, 81.4% had seen or managed a noma case, and 43.4% had referred a case to Noma Children Hospital[15].

Because of the rapidly progressing nature of the disease, a correct diagnosis is required from the entry point of care. In the experience of this group, and as reported by Farley et al. [16] and Baratti-Mayer et al. [17], approximately 30 to 40% of noma patients first consult a traditional healer. These same studies found that only 10.5% of traditional healers in Mali had some knowledge of noma [17], whereas in Sokoto state, Nigeria, a qualitative study using in-depth interviews found that traditional healers recognized attending noma patients, especially its early stages, but not knowing which disease it was [16].

In Mozambique, the national health surveillance system (Sistema de Informação de Saúde para Monitoria e Avaliação) does not currently include noma as a disease, nor have cases been reported in the scientific literature [6]. However, within a five-week period, our team was able to identify 21 noma survivors and two acute cases in Zambezia Province, evidencing its endemicity.

Nurses and medical technicians constitute the majority of the healthcare workforce in Mozambique. They undergo training for two and two and a half years [18], respectively, in *institutos técnicos médios profissionais* (pre-university level education institutions). Their curricula do not include noma, nor is it included in the medical degree program at universities. Only stomatologists and maxillofacial surgeons are trained on the disease during their

residency, while paediatricians are not. However, these specialists constitute a scarce work-force, with national ratios of one stomatologist per 88,533 population, and country totals of 14 maxillofacial surgeons and four plastic surgeons. As a result, only 15% of the primary health-care centers (*Unidades Sanitárias*) provide stomatology care, 11% in Zambezia [19]. Finally, traditional healers, who provide traditional care for ailments, vastly outnumber medical professionals, as for every physician, there are 50 traditional healers [20].

The main objective of this study was to evaluate the degree of knowledge on noma among health professionals working in health facilities in Zambezia Province, central Mozambique, focusing on both theoretical understanding and practical management competencies.

## Methods

### Ethics approval

The study was reviewed and approved by the Comité Institucional de Bioética para Saúde da Zambezia under number 219/CIBS-Z/23. The Serviço Provincial de Saúde da Zambezia validated the study and warranted access to the health facilities of the province. Formal written consent was obtained from all participants.

### Study site

The study site was chosen based on prior informal reports of noma patients living in the province. Zambezia Province has a population of 5.1 million distributed across 22 districts [21]. The highest level of health care is provided by the Quelimane Central Hospital, situated in the capital. This is the single fourth level center in the province, which hosts the only maxillofacial surgeon in the province. There is no tertiary level hospital in Zambezia. The secondary level of care is provided by the Quelimane General Hospital, which also provides some specialized care including basic surgery, and District or Rural hospitals which are situated at the main town of each district. The primary level is composed of health centers and health posts, mainly served by nurses and technicians as well as a total of 470 community health workers (*Agentes Polivalentes de Saúde*) providing home-based care in the province [18]. There are 6.2 nurses, doctors, midwives, stomatologists or pharmacists per 10,000 population in Zambezia [22], which is far below the WHO minimum threshold density of 22.8 health professionals per 10,000 population [23]. In 2016, 18.9% of the medical workforce were maternal and child health nurses, 31% generalist nurses, 14.8% medical technicians, 10% public health technicians, 7.4% laboratorians, 6.8% pharmacists, 3.1% physicians, 1.5% stomatologists, and 6.4% other technicians [18,22].

### Procedures

We conducted a cross-sectional study between January and February 2024. This was one of the objectives of the "Noma Echoes" project, performed in collaboration with the Provincial Health Service of Zambezia, aiming to provide the first scientific evidence for the presence of noma in Mozambique. Harnessing the visits to 13 districts in Zambezia looking for noma survivors, we visited health facilities on a convenience sampling basis. One to three health providers per health facility were invited to participate in the study, chosen on a quota sampling basis aiming at having a profession-diverse sample.

After signing an informed consent, a KAP questionnaire including closed and open-ended questions was read out-loud in Portuguese individually for each participant and filled-in by a trained researcher. Finally, healthcare workers were informed on the diagnosis, treatment and prevention of noma using a poster that was left at their disposal along with a phone number to encourage future reporting of suspected cases.

The KAP questionnaire (see Supplementary Material) was developed based on those previously used in Zambia [11], Burkina Faso [12], and by *Médecins Sans Frontières* in Sokoto State, Nigeria (personal communication). It contained close-ended questions regarding demographics, practices related to oral health, theoretical knowledge on noma and attitudes towards receiving noma training. Practice or management competence was evaluated with open-ended questions regarding diagnosis, treatment and recommendations for three noma practical cases, that were equal to simulated cases A, B and D in Ahlgren et al. questionnaire [11].

**Statistical analysis.** Data was digitalized independently by two coders using the Double Data Entry tool from REDCap [24,25] and compared for inconsistences by a third person. Tables and calculations were produced with R software version 4.3.3. Categorical variables were reported as frequencies and percentages, and parametric continuous variables as means with standard deviations (SD) and non-parametric continuous variables as medians with inter-quartile range (IQR).

Practical cases were graded out of 3 marks and questions on theoretical knowledge of noma were graded out of 12.4 total marks, as indicated in the mark scheme provided as Supplementary Material. Marks were further categorised into "optimal" (>=75% of total mark), "good" (50 to 74%), "suboptimal" (25 to 49%) and "very low" (<25%).

To assess differences between educational levels, secondary education level category was removed because it was represented by a single participant. Owing to low frequencies, we conducted Fisher's exact test to test differences in categorical variables between educational levels. For numerical variables, we first conducted the Shapiro–Wilk test to check normality. When data were not-normally distributed, we performed the Mann–Whitney U test. For the single case where data was normally distributed, we checked its variance and since it was equal, we computed a T-test.

All tests performed are reported.

## Results

### Characteristics of study participants

A total of 41 health providers participated in the study, working in 23 different health facilities from 12 different districts in Zambezia (Alto Molocué, Gurué, Ile, Maganda da Costa, Mocuba, Mocubela, Mopeia, Morrumbala, Namacurra, Nicoadala, Pebane and Quelimane) (Fig 1). All professionals invited to participate accepted except for one. Forty-one percent (17/41) worked at quaternary or secondary-level hospitals and 59% (24/41) at primary care centres. One had a secondary-level degree, 71% (29/41) a pre-university level degree and 27% a university degree, among which seven were stomatologists, three were physicians and one was a nutritionist (Table 1).

According to their self-reports, 32% of participants attended less than five children per day, 28% five to ten, 2% 11 to 15, 10% 16 to 24 and 20% over 24. These proportions were not statistically significant between education levels, although university-level professionals attended fewer patients per day than pre-university level. Most participants worked in health facilities where less than 50 children under five years sought oral care in a month. The median number of children attending with oral wounds per health provider and year was 10 (IQR = 23), while cheek swelling was 5.5 (IQR = 17.75) and necrosis of facial tissues was 0 (IQR = 1). Pre-university-level professionals saw significantly more children with oral wounds yearly than pre-university level professionals (p<.05).

**Practices.** The participants were asked for their oral examination practices when attending certain conditions. Among those attending children, when presented with a child

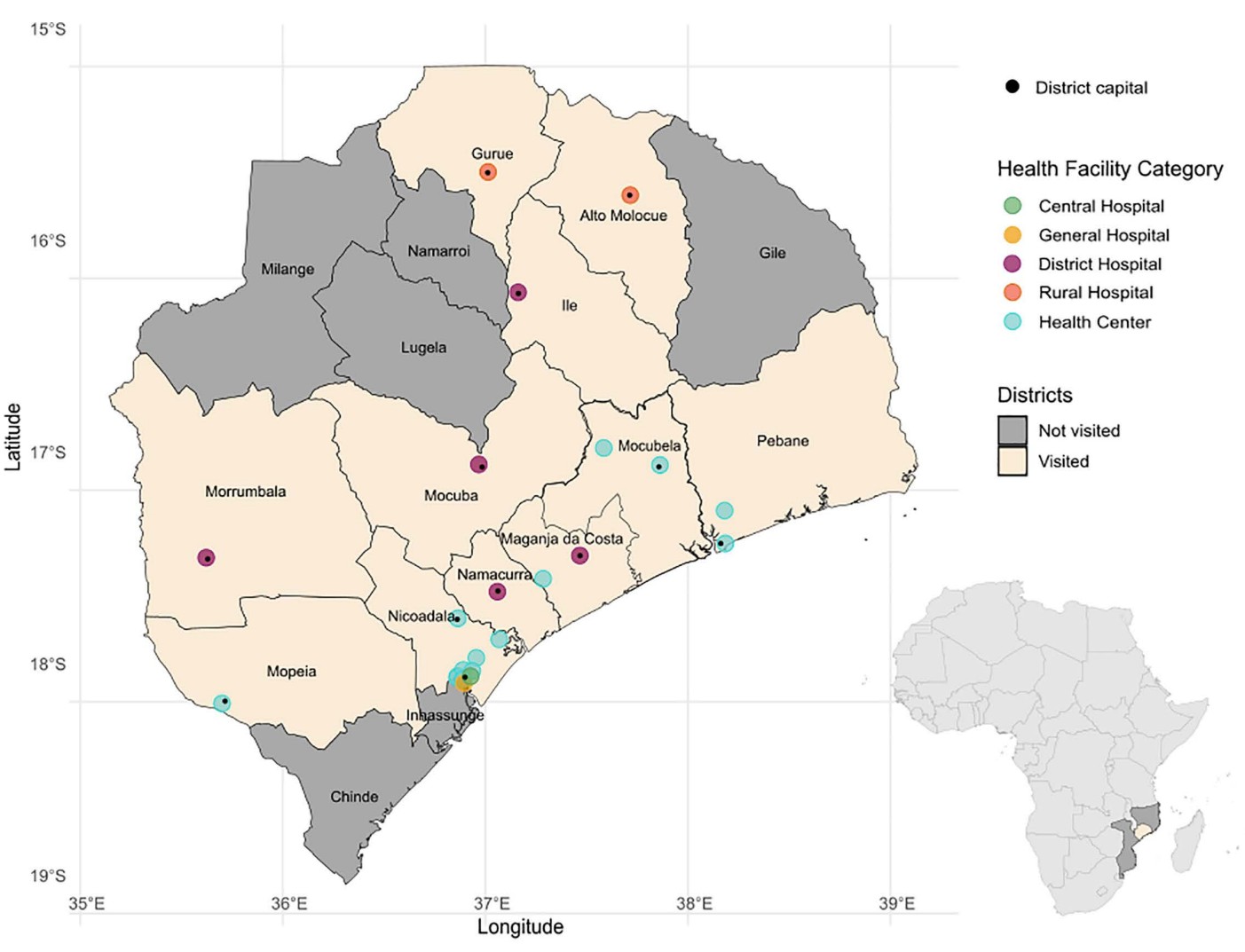

**Fig 1. Map of study site.** Districts visited are depicted in ivory and their capitals labelled as black dots. Coloured-dots indicate different geolocalisation and categories of health facilities visited. Maps' base layers were built upon shape files from the Humanitarian Data Exchange (https://data.humdata.org/dataset/cod-ab-moz), licensed under the Creative Commons Attribution for Intergovernmental Organisations (CC BY-IGO: https://creativecommons.org/licenses/by/3.0/igo/legalcode).

with malaria, 61% (22/36) said they would check their mouths, 31% (11/36) said they would not, and three were unsure. When attending to a malnourished child, 80% (28/35) declared checking their mouths, while 20% (7/35) did not. Finally, when attending to a child with HIV, 89% (31/35) said they would check their mouths, versus 9% (3/35) who said they would not and one who was unsure. These proportions were not significantly different between pre-university and university level, although they were consistently higher in the latter group.

## Knowledge

**Management competences.** All participants were presented with three practical cases before any mention of noma was made, and were asked open-ended questions regarding their diagnosis, treatment and advice (see questionnaire and mark scheme in Supplementary Materials). The first case was described as 'A mother seeks care for her two-year-old child who has gum bleeding and bad breath' and showed an image of the child's primary incisors

**Table 1. Characteristics of health providers participating in the study.**

| Variable | N* | % or median (IQR) |
|---|---|---|
| **Gender** | **41** | |
| Female | 21 | 51% |
| Male | 20 | 49% |
| **Type of health facility** | **41** | |
| Primary health centre (*Unidades Sanitárias*) | 24 | 59% |
| General Hospital (secondary level) | 3 | 7% |
| District Hospital (secondary level) | 11 | 27% |
| Rural Hospital (secondary level) | 2 | 5% |
| Central Hospital (quaternary level) | 1 | 2% |
| **Type of healthcare worker** | **41** | |
| **Secondary level** | **1** | **2%** |
| Nurse assistant (*Agente de medicina*) | 1 | 2% |
| **Pre-university level** | **29** | **71%** |
| Nutrition technician | 1 | 2% |
| Stomatology technician | 5 | 12% |
| Medical technician | 11 | 27% |
| Nurse | 5 | 12% |
| Maternal and child health nurse (midwives) | 6 | 15% |
| Stomatology nurse student | 1 | 2% |
| **University level** | **11** | **27%** |
| Nutritionist | 1 | 2% |
| Physician | 3 | 7% |
| Stomatologist (dentist) | 7 | 17% |
| **Number of children under 5 years attended per day** | **40** | |
| < 5 | 13 | 32% |
| 5 – 10 | 11 | 28% |
| 11 – 15 | 1 | 2% |
| 16 - 24 | 4 | 10% |
| > 24 | 8 | 20% |
| I don't know | 3 | 8% |
| **Number of children under 5 years seeking oral care at their health facility per month** | **38** | |
| < 26 | 22 | 58% |
| 26 – 50 | 5 | 13% |
| 51 – 75 | 3 | 8% |
| 76 – 99 | 2 | 5% |
| > 99 | 1 | 3% |
| I don't know | 5 | 13% |
| **Number of children attended by the professional per year with bleeding, swelling or gum ulceration** | **32** | 10 (23) |
| **Number of children attended by the professional per year with facial oedema** | **34** | 5.5 (17.75) |
| **Number of children attended by the professional per year with necrosis of facial tissues** | **30** | 0 (1) |
| **Ever attended or saw a patient of noma** | **41** | |
| Yes | 23 | 56% |
| No | 18 | 41% |

*Total participants who could provide the information and were applicable to respond.

with visible bleeding and inflammation of the buccal gingiva, loss of interdental papillae and some grey necrotic tissue, exhibiting signs of acute necrotizing ulcerative gingivitis (ANUG or noma stage 1). Twelve percent correctly diagnosed the case as necrotizing gingivitis or acute necrotizing gingivitis or ulcerative gingivitis. Incorrect diagnoses included simple gingivitis (mentioned by 22 participants out of 41), caries (5/41), candidiasis infection (4/41), aphthae (2/41), tonsillitis (2/41), and others mentioned smallpox, Kaposi sarcoma, herpes, abscess, HIV or malaria; and four did not know. When asked what treatment they would give, 61% said they would give amoxicillin and 68% would give analgesics; while only 7% mentioned metronidazole, 15% wound debridement and 20% nutritional support. Seven mentioned they would administer penicillin, and seven mentioned nystatin, an antifungal. Oral hygiene was the most recommended advice, mentioned by 90% of participants, followed by salty water rinses (46% recommended them), avoidance of hot food (12%) and adherence to treatment (5%). Median treatment and advice mark for case 1 was 0.75 out of 2.25 (IQR=0.75) and the median total mark when adding the diagnosis points was 0.75 out of 3 (IQR=0.75). Health providers with a university degree level were in general more correct in their answers; 27% of them diagnosed it correctly in comparison to 7% of those with a pre-university level. Eighty-two percent of them prescribed salty water rinses, compared to only 34% of those with pre-university degrees (p<.05). Likewise, their median treatment and advice mark and total mark were 0.25 points higher, although the difference did not reach statistical significance (see Table 2).

Case 2 was described as 'A 4-year-old patient with fever and inflamed lips and cheek' and pictured a child with considerable oedema on the left cheek, reaching the commissure and suborbital region (noma stage 2). Only two participants (5%) correctly diagnosed the case as noma. Thirty-two percent incorrectly diagnosed an abscess, 22% cellulitis, and 20% mumps. Other diagnoses mentioned were anaemia, conjunctivitis, malnutrition, gingivitis, myositis, mastoiditis, caries or malaria. Similarly to case 1, 59% of participants would recommend administering amoxicillin, ampicillin or penicillin when presented with a child as described in case 2, and 68% would prescribe analgesics. Seventeen percent would administer metro-nidazole, 12% gentamicin, 20% nutritional support and 7% rehydration. Importantly, only 24% would refer to another centre and 12% would advise the family to adhere to treatment. Median treatment and advice mark was 0.5 out of 2.25 (IQR=0.25) and the median total mark when adding the diagnosis points was 0.5 out of 3 (IQR=0.25). Practitioners with university degrees performed better, although differences were not statistically significant except for the recommendation of giving nutritional support, which was given by 45% of professionals with a university degree, versus 10% of those at pre-university level. As in case 1, the median treatment and advice mark and total mark were 0.25 points higher for those with a university degree, although the difference did not reach statistical significance.

Case 3 image was accompanied by the description 'A 25-year-old patient with gangrenous tissue and a hole in the cheek' and showed a young woman with significant soft tissue loss, extending several centimetres from the left commissure towards the angle of the mandible. This included the lower cheek, as well as the upper and lower lip tissue distal of the premo-lars. The defect exposed the underlying dentition and oral cavity, in which all mandibular molars are missing and gingival recession can be seen around the remaining teeth. Significant raised scarring can be seen around the defect, which is most prominent around the mandible (noma stage 4). Forty-six percent correctly diagnosed noma, 79% of whom had heard about the disease previously. Among the rest, three diagnosed an ulcer, two an oral gangrene, two an abscess, one an oral cancer, one an infection and 13 did not know. Sixty-three percent recommended referring the patient to a specialist, 22% recommended administering amox-icillin, ampicillin or penicillin, 15% metronidazole, 15% gentamicin and 15% removal of

**Table 2.** Performance of participants on diagnosing and prescribing treatment and advise when presented with three practical cases of noma patients. A correct diagnosis was given 0.75 points and each treatment or advice mentioned was given 0.25 points (except referral which scored 0.5). Data is presented as totals and segregated by highest educational level. Secondary level was omitted as it was only represented by one participant. To assess differences between the two educational levels, Fisher's exact test was used for categorical variables and Mann-Whitney U test for numerical variables except those with a superscript star (*) which were tested with a T-test as they were parametric. Statistics in bold were statistically significant (p<.05).

| Variable | Total | | | Pre-university level | | | University level | | | Test statistic | P-value |
|---|---|---|---|---|---|---|---|---|---|---|---|
| | N | Median/% | IQR | N | Median/% | IQR | N | Median/% | IQR | | |
| **CASE 1** | 41 | | | 29 | | | 11 | | | | |
| **Correct diagnosis** (ANUG/ Noma stage 1/ Ulcerative gingivitis/ Necrotizing gingivitis) | 5 | 12% | | 2 | 7% | | 3 | 27% | | 4.8 | 0.12 |
| **Treatment and advise recommended** | | | | | | | | | | | |
| Amoxicillin | 25 | 61% | | 18 | 62% | | 7 | 64% | | 1.1 | 1.00 |
| Metronidazole | 3 | 7% | | 2 | 7% | | 1 | 9% | | 1.3 | 1.00 |
| Wound debridement | 6 | 15% | | 4 | 14% | | 2 | 18% | | 1.4 | 1.00 |
| Nutritional support | 8 | 20% | | 5 | 17% | | 3 | 27% | | 1.8 | 0.66 |
| Analgesics | 28 | 68% | | 19 | 66% | | 8 | 73% | | 1.4 | 1.00 |
| Oral hygiene | 37 | 90% | | 25 | 86% | | 11 | 100% | | Inf | 0.56 |
| Salty water rinses | 19 | 46% | | 10 | 34% | | 9 | 82% | | **8.1** | **0.01** |
| Adherence to treatment | 2 | 5% | | 2 | 7% | | 0 | 0% | | 0.0 | 1.00 |
| To avoid hot food | 5 | 12% | | 3 | 10% | | 2 | 18% | | 1.9 | 0.60 |
| Treatment and advice mark (out of 2.25) | 41 | 0.75 | 0.75 | 29 | 0.75 | 0.75 | 11 | 1 | 0.5 | 108.0 | 0.11 |
| **Total mark (out of 3)** | 41 | 0.75 | 0.75 | 29 | 0.75 | 0.75 | 11 | 1 | 0.875 | 97.0 | 0.06 |
| **CASE 2** | 41 | | | 29 | | | 11 | | | | |
| **Correct diagnosis** (noma/ stage 3/ noma oedema stage) | 2 | 5% | | 1 | 3% | | 1 | 9% | | 2.7 | 0.48 |
| **Treatment and advise recommended** | | | | | | | | | | | |
| Amoxicillin/Ampicillin/Penicillin | 24 | 59% | | 16 | 55% | | 7 | 64% | | 1.4 | 0.73 |
| Metronidazole | 7 | 17% | | 3 | 10% | | 4 | 36% | | 4.7 | 0.08 |
| Gentamicin | 5 | 12% | | 3 | 10% | | 2 | 18% | | 1.9 | 0.60 |
| Nutritional support | 8 | 20% | | 3 | 10% | | 5 | 45% | | **6.8** | **0.02** |
| Rehydration | 3 | 7% | | 3 | 10% | | 0 | 0% | | 0.0 | 0.55 |
| Analgesics | 28 | 68% | | 21 | 72% | | 6 | 55% | | 0.5 | 0.45 |
| Referral | 10 | 24% | | 7 | 24% | | 3 | 27% | | 1.2 | 1.00 |
| Adherence to treatment | 5 | 12% | | 5 | 17% | | 0 | 0% | | 0.0 | 0.30 |
| Treatment and advice mark (out of 2.25) | 41 | 0.5 | 0.25 | 29 | 0.5 | 0.25 | 11 | 0.75 | 0.5 | 139.0 | 0.53 |
| **Total mark (out of 3)** | 41 | 0.5 | 0.25 | 9 | 0.5 | 0.25 | 11 | 0.75 | 0.5 | 139.5 | 0.54 |
| **CASE 3** | 41 | | | 29 | | | 11 | | | | |
| **Diagnosis** (noma/ stage 4/ noma scarring stage) | 19 | 46% | | 10 | 34% | | 9 | 82% | | **8.1** | **0.01** |
| **Treatment and advise recommended** | | | | | | | | | | | |
| Removal of necrotic tissue or wound debridement | 6 | 15% | | 3 | 10% | | 3 | 27% | | 31.4 | 0.32 |
| Referral | 26 | 63% | | 15 | 52% | | 11 | 100% | | **inf** | **0.00** |
| Amoxicillin/Ampicillin/Penicillin | 9 | 22% | | 6 | 21% | | 3 | 27% | | 1.4 | 0.69 |
| Metronidazole | 6 | 15% | | 4 | 14% | | 2 | 18% | | 1.4 | 1.00 |
| Gentamicin | 6 | 15% | | 5 | 17% | | 1 | 9% | | 0.5 | 1.00 |
| Adherence to treatment | 4 | 10% | | 3 | 10% | | 1 | 9% | | 0.9 | 1.00 |
| Wound dressing | 15 | 37% | | 14 | 48% | | 1 | 9% | | **0.1** | **0.03** |
| Nutritional support | 5 | 12% | | 1 | 3% | | 4 | 36% | | **14.6** | **0.02** |
| Regular visits to health facility | 2 | 5% | | 2 | 7% | | 0 | 0% | | 0.0 | 1.00 |

*(Continued)*

**Table 2.** (Continued)

| Variable | Total | | | Pre-university level | | | University level | | | | |
|---|---|---|---|---|---|---|---|---|---|---|---|
| | N | Median/% | IQR | N | Median/% | IQR | N | Median/% | IQR | Test statistic | P-value |
| Treatment and advice mark (out of 2.25) | 41 | 0.5 | 0.5 | 29 | 0.25 | 0.5 | 11 | 0.5 | 0.625 | 125.0 | 0.28 |
| **Total mark (out of 3)** | 41 | 1 | 1 | 29 | 0.75 | 0.75 | 11 | 1.25 | 0.65 | **80.0** | **0.02** |
| **Total management competence mark (out of 9)** | 41 | **2.25** | **1.5** | 29 | **2** | **1** | 11 | **3.25** | **1.375** | 85.5 | **0.03** |

necrotic tissue. Regarding advise to the patient and their family, 37% recommended going regularly to the health facility for wound dressing, 12% advised on nutrition, 10% on adhering to treatment and 5% on visiting regularly health facilities. The median treatment and advise mark were 0.5 out of 2.25 (IQR=0.5) and the median total mark when adding the diagnosis points was 1 out of 3 (IQR=1). Eighty-two percent of practitioners with a university degree correctly diagnosed noma, while only 34% of those with pre-university degrees did so (p<.05). Similarly, university-level professionals recommended in a significantly higher proportion, the referral of the patient (100% vs 52%, p<.05), and advised on nutrition (36% vs 3%, p<.05). On the contrary, pre-university level professionals performed slightly better in some treatment recommendations such as giving gentamicin, advising regular health facility visits , and wound dressing (48% vs 9%, p<.05). The total mark for the third case was significantly higher in the university-level professionals, who scored 1.25 out of 3 (IQR=0.65) versus 0.75 out of 3 (IQR=0.75) in pre-university level professionals (Table 2).

When adding up the scores of the three practical cases, median score was 2.25 out of 9 (SD=2.1). Overall, university-level practitioners had a significantly better management competence than pre-university level, with a mean score of 3.25 out of 9, versus 2 out of 9 (p<.05) (Table 2). The best management competence was found among stomatology technicians (mean mark 3.5), followed by stomatologists (mean 3.4), physicians (mean 3.2), medical technicians (mean 2.2), nutrition technicians (mean 2), nurses (mean 1.7), midwives (mean 1.3) and lastly nutritionists and community health workers (mean 1).

When categorising their marks in quartiles, among pre-university practitioners, the vast majority (72%) had a "very low" management competence, 24% "suboptimal" and 3% "good"; while the majority of university-level professionals had a "suboptimal" management competence (64%), 18% "good", and 18% "very low". None had an "optimal" management competence.

**Theoretical knowledge on noma.** After the practical cases, participants were asked whether they had ever heard about noma. Fifty-nine percent (24/41) were aware of noma, 52% among the pre-university level professionals (mainly nurses, medical technicians and stomatology technicians), and 82% among the university-level ones (Table 3). Seventy-one percent had heard about it during their studies (as a medical technician, stomatology technician, physician, nurse or stomatologist), 12.5% from the media, 21% at work and one during a course on HIV.

Further close-ended questions on theoretical knowledge of noma were asked to the 24 participants who responded affirmatively. All of them knew that noma affects the orofacial region, 83% knew it can be prevented, 75% that it is not contagious and 67% that it is a bacterial infection. However, only 14% could correctly order the noma stages (most common mistake being ordering oedema before ANUG), 17% knew its mortality rate and 12% that it evolves in less than two weeks (Table 3). Regarding risk factors, all of them recognized bad oral hygiene as a risk factor, 96% malnutrition, 83% smoking and 67% correctly discarded high blood pressure as a risk factor. Likewise, performance was relatively high in identifying

**Table 3. Performance of participants on theoretical knowledge of noma.** Data is presented as totals and segregated by highest educational level. Secondary level was omitted as it was only represented by one participant. To assess differences between the two educational levels, Fisher's exact test was used for categorical variables and T-test for numerical variables as they were parametric (#). Statistics in bold were statistically significant (p<.05). Marks for each question are indicated in the mark scheme in annex.

| Variable | Total | | | Pre-university level | | | University level | | | Test statistic | P-value |
|---|---|---|---|---|---|---|---|---|---|---|---|
| | N | Mean/% | SD | N | Median/% | SD | N | Mean/% | SD | | |
| **Have you heard about noma before?** | **41** | | | **29** | | | **11** | | | 4.06 | 0.15 |
| Yes | 24 | 59% | | 15 | 52% | | 9 | 82% | | | |
| No | 17 | 41% | | 14 | 48% | | 2 | 18% | | | |
| **Which region is affected by noma?** | **24** | | | **15** | | | **9** | | | | |
| Orofacial region | 24 | 100% | | 15 | 100% | | 9 | 100% | | | |
| **Can you order noma stages according to WHO?** | **24** | | | **15** | | | **9** | | | 3.76 | 0.53 |
| Correctly ordered | 3 | 12.5% | | 1 | 7% | | 2 | 22% | | | |
| Not correctly ordered | 21 | 87.5% | | 14 | 93% | | 7 | 78% | | | |
| **Can noma be prevented?** | **24** | | | **15** | | | **9** | | | 1.95 | 1.00 |
| Yes | 20 | 83% | | 12 | 80% | | 8 | 89% | | | |
| I don't know | 4 | 17% | | 3 | 20% | | 1 | 11% | | | |
| **Which is the mortality of noma?** | **24** | | | **15** | | | **9** | | | 1.81 | 0.61 |
| 10%* | 7 | 29% | | 5 | 33% | | 2 | 22% | | | |
| 90% | 4 | 17% | | 2 | 13% | | 2 | 22% | | | |
| I don't know* | 13 | 54% | | 8 | 53% | | 5 | 56% | | | |
| **How long does noma take to evolve from gingivitis to gangrene?** | **24** | | | **15** | | | **9** | | | Inf | **0.04** |
| Less than 2 weeks | 3 | 12% | | 0 | 0% | | 3 | 33% | | | |
| 3 months* | 7 | 29% | | 4 | 27% | | 3 | 33% | | | |
| A year* | 5 | 21% | | 5 | 33% | | 0 | 0% | | | |
| I don't know* | 9 | 38% | | 6 | 40% | | 3 | 33% | | | |
| **Is malnutrition a risk factor for noma?** | **24** | | | **15** | | | **9** | | | Inf | 1.00 |
| Yes | 23 | 96% | | 14 | 93% | | 9 | 100% | | | |
| I don't know | 1 | 4% | | 1 | 7% | | 0 | 0% | | | |
| **Is high blood pressure a risk factor for noma?** | **24** | | | **15** | | | **9** | | | 2.25 | 0.66 |
| Yes* | 3 | 12% | | 2 | 13% | | 1 | 11% | | | |
| No | 16 | 67% | | 9 | 60% | | 7 | 78% | | | |
| I don't know* | 5 | 21% | | 4 | 27% | | 1 | 11% | | | |
| **Is bad oral hygiene a risk factor for noma?** | **24** | | | **15** | | | **9** | | | | |
| Yes | 24 | 100% | | 15 | 100% | | 9 | 100% | | | |
| **Is smoking a risk factor for noma?** | **24** | | | **15** | | | **9** | | | 0.55 | 0.61 |
| Yes | 20 | 83% | | 13 | 87% | | 7 | 78% | | | |
| No* | 2 | 8% | | 1 | 7% | | 1 | 11% | | | |
| I don't know* | 2 | 8% | | 1 | 7% | | 1 | 11% | | | |
| **What is noma aetiology?** | **24** | | | **15** | | | **9** | | | inf | 0.51 |
| Bacteria* | 16 | 67% | | 10 | 67% | | 6 | 67% | | | |
| Unknown* | 6 | 25% | | 3 | 20% | | 3 | 33% | | | |
| Virus | 2 | 8% | | 2 | 13% | | 0 | 0% | | | |
| **Is noma contagious?** | **24** | | | **15** | | | **9** | | | 3.80 | 0.35 |
| Yes* | 2 | 8% | | 1 | 7% | | 1 | 11% | | | |
| No | 18 | 75% | | 10 | 67% | | 8 | 89% | | | |
| I don't know* | 4 | 17% | | 4 | 27% | | 0 | 0% | | | |

*(Continued)*

**Table 3.** (Continued)

| Variable | Total | | | Pre-university level | | | University level | | | | |
|---|---|---|---|---|---|---|---|---|---|---|---|
| | N | Mean/% | SD | N | Median/% | SD | N | Mean/% | SD | Test statistic | P-value |
| **Total theoretical knowledge mark** (Out of 12.4) | 24 | 8.75 | 2 | 15 | 8.2 | 1.85 | 9 | 9.1 | 0.7 | -2.153[#] | 0.04 |
| **Total knowledge mark** (management competence mark + theoretical knowledge mark) (Out of 21.4) | 24 | 12 | 2.1 | 15 | 11 | 1.8 | 9 | 13 | 1.8 | -2.705[#] | 0.01 |

*indicate responses that were considered as a single category (either correct or incorrect) for the statistical test to assess differences in performance.

preventive actions for the disease, with 100% recognizing improvement of oral hygiene, 96% improving nutritional status and 62% recognizing childhood vaccination as a preventive measure. Most participants correctly recognized mouthwashes (78%), disinfection of the wound (96%), antibiotics (96%) and vitamin supplementation (96%) as treatment for noma. Finally, 96% recognized improving access to healthcare as a good public health measure to prevent noma, 88% to improve life conditions, 75% to improve referral systems between health centres and 58% to partner with traditional healers.

When comparing between education level, those with a university degree answered slightly better in most questions, and significantly outperformed their pre-university colleagues in knowing the time progression of noma, as 33% correctly answered that it evolves within less than two weeks while none of the pre-university level knew of its rapidlyprogressing nature (p<.05). The median score on knowledge of noma for the 24 participants who had previously heard about the disease was 8.75 out of 12.4 (SD=2), 9.1 in university degree level professionals and 8.2 in pre-university degree professionals (p<.05). When classifying their marks in quartiles, among pre-university practitioners, the vast majority (73%) had a "good" theoretical knowledge, 20% an "optimal" one and 7% "suboptimal". Among university-level professionals, 56% had a "good" theoretical knowledge, and 44% an "optimal" one.

When adding up the scores of the practical cases and theoretical knowledge on noma, among those who had heard about noma before, and therefore were applicable to respond to the theoretical knowledge questions, mean mark was 12 out of 21.4 (SD=2.1). University-level practitioners did significantly better than pre-university level, with a mean mark of 13 out of 21.4, versus 11 out of 21.4 (p<.05) (Table 3). Physicians scored the highest with a mean mark of 14, followed by stomatologists (mean 13), stomatology technicians (mean 12), nurses (mean 10) and medical technicians (mean 9.9). The rest of professional categories' participants had not heard about noma before. Looking at quartile marks, among pre-university practitioners, 60% had a "suboptimal" total score and 40% "good"; while the majority of university-level professionals had a "good" score (78%), one had "suboptimal" score, and one an "optimal" score.

After having answered all questions, practitioners were explained noma's signs and symptoms and the recommended treatment. Out of all participants, 56% (23/41) had seen a case of noma before. The question was asked broadly enough to include having encountered a noma survivor in the street or as a patient. Eleven (26.8%) had attended to the patients themselves during the acute phase. Among them, four were physicians, two midwives, two stomatology technicians, a nurse, a stomatologist and a medical technician. In the northern districts of Gilé and Alto Molocué, two physicians mentioned having attended to three-year-old children with acute noma in November and April 2022, respectively. A stomatology technician in Mocuba's Hospital recalled having seen at least 12 cases, the last being in 2014, while in 2008 and 2009 it was frequent, with a new case being admitted every three months. In Morrumbala's District

Hospital a physician recalled having attended three acute cases simultaneously at Nampula's Hospital around 2017.

All participants declared having antibiotics at their disposal, although shortages were mentioned by several of them. Eighty-two percent had access to wound debridement and 77% to food supplements. All the participants who reported a lack of wound debridement equipment came from primary health centres, and out of the seven participants reporting a lack of vitamins, six came from primary healthcare centres, with one participant from a rural hospital.

All participants were interested in receiving further training on noma, and the preferred methods were weekend courses administered by specialists (63%) and being involved in research projects (59%). The least preferred were having data on local studies on noma (32%) and online courses (39%).

## Discussion

We evaluated the knowledge, attitudes and practices of health practitioners in quaternary, secondary and primary care levels in Zambezia province, central Mozambique. Although over half of them had heard about the disease before, few of those knew its elevated mortality rate or its rapid progression, or could diagnose or treat appropriately at its early stages. Practitioners with university degrees had a generally higher level of knowledge on noma specificities and management competences than practitioners with pre-university level, which constitutes the largest share of the healthcare workforce in the region.

Routine oral screening for children at health facilities has been suggested as a key prevention strategy for noma [1]. In our study, the reported frequency of oral screening in patients with conditions predisposing to noma was relatively high, with an over 80% screening rate for patients with HIV or malnutrition, and 61% with malaria. These are comparable to findings by Brattström in Burkina Faso [12] and by Ahlgren in Zambia [11]. However, the self-reported median number of children attending with gingivitis was 10 per year per participant. Given the estimated 7% prevalence of bleeding gums among children under 12 at national level [26], if screenings were effectively as high, a larger number of gingivitis cases would be expected. For example, a professional seeing 10 children per day would attend around 2,300 children per year; assuming a 60% oral screening rate and a 7% gingivitis prevalence, 97 cases of gingivitis would have been detected by that professional, while the reported median was 10. This discrepancy suggests a possible response bias, where health practitioners may have reported what they believed to be the correct practice rather than their actual routine behaviour. In any case, these are self-reported figures, and therefore they are subject to recall bias.

Surprisingly, 59% of practitioners in our study had heard of noma, which is comparable to the participants in the Zambian study [11] around 54%, but much lower than the 91% rate reported in Burkina Faso [12], or the 78.8% to 83.7% in North West Nigeria [14,15]. However, 34% of participants in Burkina Faso had previously participated in a noma training; and therefore, percentage in general health community might be lower. On the contrary, in North West Nigeria, the studies were conducted five kilometres away from Noma Children Hospital, where many awareness campaigns and research have been conducted for several decades [13], and therefore, we would expect higher level of knowledge than in Mozambique, where to our knowledge, no specific training on noma has been conducted yet.

Even though the majority of participants had previously heard about the disease, when presented with practical cases of noma, only 12% of participants could correctly diagnose noma at stage 1 (ANUG) and 5% at stage 2 (oedema), which are still reversible, although only if a diagnosis is made and appropriate treatment is administered. More participants (46%) could diagnose noma at stage 4 (scarring), 79% among those who had heard about the disease

previously. In Zambia, all participants had pre-university education level, and their results were comparable to those of our Mozambican subsample with the same education level [11]. While the Zambian participants demonstrated a higher rate of correctness in diagnosing stage 1 (71%), this was largely due to their inclusion of simple gingivitis as a correct answer, accounting for 69% of responses [11]. Using WHO definition for stage 1 [2], however, Mozambican participants outperformed the Zambian group (7% vs 2% correctness). For stage 2 and stage 4 cases, diagnosis correctness was equally evaluated in both studies and results were comparable (6% in stage 2 for Zambian participants versus 3% in our study, and 29% in stage 4 in Zambia compared to 34% in Mozambique). The very low diagnosis correctness for stage 2 can be partially explained by the fact that the signs and symptoms described in the practical case were not exclusive of noma.

Despite erring in the diagnosis, approximately 60% of participants recommended amoxicillin for treating stages 1 and 2, but fewer than 20% mentioned metronidazole and gentamicin, consistent with findings from the Zambian study [11]. Nutritional and rehydration support, critical aspects of noma care, were the most frequently overlooked treatments, together with referral, which becomes crucial from stage 2 onward due to the rapid progression of the disease to necrosis. Only 24% of practitioners would have referred a child with oedema stage to a higher-level health facility. At the later scarring stage, when the acute stage of disease is already concluded, 52% of the pre-university level participants advised referral, which is right below the 70 to 100% referral rates reported by nurses in Burkina Faso [12]. Among pre-university level participants, 52% recommended referral in Zambezia, compared to 31% referral rate reported by the Zambian study [11]. Total management competence was deemed "very low" (<25%) for 72% of the pre-university study participants, which contrasts with the majority of Burkina Faso's nurse participants having a "suboptimal" score (25-49%) with only 8% having a "very low" score [12]. However, their scoring system is not clearly detailed, and the difference could be due to a laxer scoring system leading to an apparent better management competence.

Theoretical knowledge on the disease among participants who had previously heard about noma was apparently better than management competence, however this might be partially due to being assessed by close-ended questions with available answers, as opposed to management competence with was evaluated by open-ended questions. In the Zambian study, theoretical knowledge was assessed with open-ended questions and the general knowledge level was categorised as "very low" in 66% of participants [11], contrasting with 55% of Burkina Faso's nurses [12] and 93% of pre-university professionals in Mozambique scoring "good" or "optimal". It is noteworthy that, despite Sokoto being an epicentre for noma research and treatment for decades, health workers in Zambezia outperformed their Sokoto counterparts on certain questions. In particular, fewer participants in Sokoto identified poor oral hygiene as a risk factor for noma (68.3% vs 100%), or recognised the use of antibiotics as treatment (37.4% vs 96%) [14]. Future studies in both regions should address these variations and strive to generate more robust and comparable evidence.

Notably, 26.8% of participants reported having personally attended an acute noma patient during their careers, a figure that contrasts with the 7% reported in Burkina Faso [12], and the 11% in Zambia [11]. A possible explanation for this difference is that nearly half of those reporting having attended an acute patient in our study were physicians or stomatologists, whereas the Burkina Faso and Zambia studies primarily included nurses and healthcare workers, respectively. We hypothesise that physicians are more likely to have worked in referral hospitals covering large catchment areas, where severe conditions are transferred to.

In Mozambique, the first entry point to care for most rural population are traditional healers, community health workers (*agentes polivalentes*), or medical technicians

at primary care centres. However, and as evidenced by their better performance in this study, noma training currently targets oral health specialists, whose numbers are critically low. Some medical technicians, nurses and physicians had heard about noma during their studies, despite it not being officially included in their curricula, but their management competence was very low. To improve early detection and management, traditional healers and primary care workers must be trained to recognise its early signs and to refer patients when needed. Moreover, physicians, paediatricians, dentists, and nutritionists must know how to manage acute cases, while physiotherapists, psychologists and maxillofacial and plastic surgeons are essential for addressing noma's long-term functional and psychological sequelae.

As observed in our study, and corroborated in other countries, traditional healers [16,17], and primary care workers express willingness to receive training on noma [12,15]. Key messages can be effectively transmitted through short training programs to nurses [12], and to traditional healers [27], as demonstrated in Burkina Faso. Implementing such programs is not only imperative from the point of view of warranting the human right to health [28], but also to avoid the indirect costs from premature deaths, amounting to billions at the national level [29].

The primary limitation of this study lies in its restricted representativeness. Due to resource constraints, sampling was done on a convenience basis and the sample size was small. The majority of the health facilities visited were also located near urban areas, which may not accurately reflect the rural ones. Additionally, we did not include community health workers or traditional healers, both of whom are crucial for early diagnosis of noma. Another limitation was that  although care was taken by the team to avoid any mention of noma prior to the questionnaire, participants might have realised the topic of the study beforehand, therefore potentially biasing diagnostic answers in the management competence section. We also did not assess the management competence for noma stage 3, and the practical case used to evaluate noma stage 2 was not sufficiently specific, hindering the reliability of the results. Lastly, figures on the number of patients attended are subject to recall bias, and should be contrasted with routine health data in future studies.

## Conclusions

The prevalence of noma in Mozambique remains undetermined; however, 56% of participants in this study reported having encountered a noma patient in their practice or personal lives. Noma treatment was readily available in the health facilities visited, as well as the possibility for referral. However, most health professionals could not identify early stages of noma, nor knew its high mortality, its rapid progression, or its correct management. It is therefore not a matter of lack of access to treatment, but lack of awareness about the disease, which hinders timely diagnosis and appropriate care. It is urgent to implement comprehensive training programs across all levels of healthcare providers to ensure the fundamental right to health. Implementation research studies which incorporate the perspectives of noma survivors and their communitieswill be key to identify the most effective strategies for training and awareness raising within local healthcare and sociocultural contexts.

## Supporting information

**S1 File.  Questionnaire** .
(PDF)

**S2 File.  Manuscript in Portuguese.**
(PDF)

## Acknowledgments

We would like to thank the health practitioners who took the time to answer to our questions and learn about noma. We thank Elise Farley, who kindly reviewed our questionnaire. We would also like to thank the heads of the hospitals visited that permitted the work to take place successfully. The assistance of ChatGPT was used to improve readability and grammar correctness of this manuscript. After using this tool, the authors reviewed and edited the content as needed and take full responsibility for the content of the publication.

## Author contributions

**Conceptualization:** Marta Ribes, Fernando Padama, Carlos Chaccour.

**Data curation:** Marta Ribes, Fizaa Halani, Romina Rios-Blanco.

**Formal analysis:** Marta Ribes, Fizaa Halani, Gemma Moncunill, Carlos Chaccour.

**Funding acquisition:** Marta Ribes, Carlos Chaccour.

**Investigation:** Marta Ribes, Abdala Atumane, Milagre Andurage, Eldo Elobolobo, Tairo Sumine, Luis Transval, Fernando Padama.

**Methodology:** Marta Ribes, Abdala Atumane, Fernando Padama, Carlos Chaccour.

**Supervision:** Gemma Moncunill, Fernando Padama, Carlos Chaccour.

**Visualization:** Marta Ribes, Eldo Elobolobo.

**Writing – original draft:** Marta Ribes.

**Writing – review & editing:** Marta Ribes, Fizaa Halani, Abdala Atumane, Milagre Andurage, Eldo Elobolobo, Gemma Moncunill, Romina Rios-Blanco, Tairo Sumine, Luis Transval, Fernando Padama, Carlos Chaccour.

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
