## [Decision Letter · Decision Letter 0]

15 Jan 2025

PNTD-D-24-01714Knowledge, Attitudes and Practices of Healthcare Workers Towards Noma in Zambezia, MozambiquePLOS Neglected Tropical DiseasesDear Dr. Ribes, Thank you for submitting your manuscript to PLOS Neglected Tropical Diseases. After careful consideration, we feel that it has merit but does not fully meet PLOS Neglected Tropical Diseases's publication criteria as it currently stands. Therefore, we invite you to submit a revised version of the manuscript that addresses the points raised during the review process. Please submit your revised manuscript within 30 days Mar 16 2025 11:59PM. If you will need more time than this to complete your revisions, please reply to this message or contact the journal office at plosntds@plos.org. Please include the following items when submitting your revised manuscript: * A rebuttal letter that responds to each point raised by the editor and reviewer(s). You should upload this letter as a separate file labeled 'Response to Reviewers '. This file does not need to include responses to any formatting updates and technical items listed in the 'Journal Requirements' section below. * A marked-up copy of your manuscript that highlights changes made to the original version. You should upload this as a separate file labeled 'Revised Manuscript with Track Changes '. * An unmarked version of your revised paper without tracked changes. You should upload this as a separate file labeled 'Manuscript '. If you would like to make changes to your financial disclosure, competing interests statement, or data availability statement, please make these updates within the submission form at the time of resubmission. Guidelines for resubmitting your figure files are available below the reviewer comments at the end of this letter. We look forward to receiving your revised manuscript.

Kind regards,

Stuart Robert AinsworthAcademic EditorPLOS Neglected Tropical Diseases Ana LTO NascimentoSection EditorPLOS Neglected Tropical Diseases Shaden Kamhawico-Editor-in-ChiefPLOS Neglected Tropical Diseasesorcid.org/0000-0003-4304-636XX

Paul Brindley

co-Editor-in-Chief

**Journal Requirements:**

At this stage, the following Authors/Authors require contributions: Marta Ribes, Fizaa Halani, Abdala Atumane, Milagre Andurage, Eldo Elobolobo, Gemma Moncunill, Romina Ríos-Blanco, Tairo Sumine, Luis Transval, Fernando Padama, and Carlos Chaccour. Please ensure that the full contributions of each author are acknowledged in the "Add/Edit/Remove Authors" section of our submission form.

- ® on page: 8.

Potential Copyright Issues:

- Please confirm that you are the photographer of Striking Image, or provide written permission from the photographer to publish the photo(s) under our CC BY 4.0 license.

- Figure 1. Please provide a direct link to the base layer of the map (i.e., the country or region border shape) and ensure this is also included in the figure legend; and provide a link to the terms of use / license information for the base layer image or shapefile. We cannot publish proprietary or copyrighted maps (e.g. Google Maps, Mapquest) and the terms of use for your map base layer must be compatible with our CC BY 4.0 license.

6) Please amend your detailed Financial Disclosure statement. This is published with the article. It must therefore be completed in full sentences and contain the exact wording you wish to be published. Please ensure that the funders and grant numbers match between the Financial Disclosure field and the Funding Information tab in your submission form. Note that the funders must be provided in the same order in both places as well.

**Reviewers' comments:**

Reviewer's Responses to Questions

**Key Review Criteria Required for Acceptance?**

**Methods**

-Are the objectives of the study clearly articulated with a clear testable hypothesis stated?

-Is the study design appropriate to address the stated objectives?

-Is the population clearly described and appropriate for the hypothesis being tested?

-Is the sample size sufficient to ensure adequate power to address the hypothesis being tested?

-Were correct statistical analysis used to support conclusions?

-Are there concerns about ethical or regulatory requirements being met?

Reviewer #1: The methods are clearly described. see general comments for minor points.

Reviewer #2: Methods are clearly presented and well-structured. I found the “Study site”paragraph highly informative for understanding the context in which the research was conducted, and I have no concerns regarding ethical requirements.

However, the objectives of the study are not clearly outlined in the Methods section. I suggest incorporating them at the beginning of the "Procedures" paragraph (end of line 163). Alternatively, consider presenting them more comprehensively at the end of the Introduction (lines 144–145). Currently, only the main objective (knowledge on noma) is mentioned (lines 144-145), while the secondary objectives (assessment of practices related to oral health and attitudes toward noma training) are not addressed.

**Results**

-Does the analysis presented match the analysis plan?

-Are the results clearly and completely presented?

-Are the figures (Tables, Images) of sufficient quality for clarity?

Reviewer #1: The results are clearly described. See general comments for minor points.

Reviewer #2: The results are clear and comprehensive, with the tables and figures provided effectively supporting their interpretation. However, In some sections, the data presentation is overly detailed. To make the text more concise and easier to read, the authors could consider omitting some of the data described in the Results, directing readers to the tables for additional details—particularly in the subsection “Management competences”.

Besides, I would like to raise the following concerns regarding lines 232–241:

- I suggest moving lines 232–241 to the subsection titled “Specific knowledge on noma,” placing them after line 354 (“...21% at work and one during a course on HIV”). This adjustment would allow the authors to first report how many professionals are aware of noma and then follow with how many have actually encountered cases.

- Lines 236–241 seem to add limited value to the subject due to their anecdotal nature. I suggest deleting these lines and simply stating that the professionals who encountered noma cases were from different geographical areas, with the number of cases per professional ranging from 1 to 12.

**Conclusions**

-Are the conclusions supported by the data presented?

-Are the limitations of analysis clearly described?

-Do the authors discuss how these data can be helpful to advance our understanding of the topic under study?

-Is public health relevance addressed?

Reviewer #1: The conclusion section needs to be shortened and focused on the key insights from the study.

Reviewer #2: The article is thoroughly discussed and provides an effective critical analysis of its limitations and biases. The conclusions are well-situated within the context of existing literature.

I believe the need to train healthcare personnel on noma is a crucial aspect of the article, given its significant practical implications for public health. However, this topic is mentioned only in the Conclusions rather than being explored in the Discussion. Thus, I suggest revising as follows:

- Move lines 496–502 (“In Mozambique, the first entry point… as demonstrated in Burkina Faso”) to the end of the Discussion (after line 485). Follow this by adding lines 504–508 (“Traditional healers and primary care workers… and long-term sequelae”). Consider expanding the discussion on the topic with practical examples of how training has been implemented in other settings or by referencing experiences beyond Burkina Faso.

- Keep lines 503–504 and 508–509 in the Conclusions (following line 495): “It is therefore not a matter of lack of access to treatment, but lack of access to timely and appropriate care. It is crucial to implement comprehensive training programs across all levels of healthcare providers. Implementation research studies which incorporate the perspectives of noma survivors and their communities, will be key to identify the most effective strategies for training and awareness raising within local healthcare and sociocultural contexts.” Consider adding further details on why educating healthcare personnel on noma is so important by highlighting economic and social implications (eg., preventing disabilities and psychological impacts on children; avoiding the need for complex reconstructive surgeries).

**Editorial and Data Presentation Modifications?**

Reviewer #1: See general comments.

Reviewer #2: Minor revisions:

Line 31: spelling mistake “sixty-three" inestead of "tree"

Line 176-7, “…Médecins Sans Frontières in Sokoto State, Nigeria” there’s a missing quote

Line 222: as stated in Table 1, authors should replace “34” with “24”: “11% 16 to 24 and 22% over 24”

Line 223: I suggest adding “attended”, instead of “tended to”: “…university-level professionals attended fewer patients”

Line 324: According to Table 2, “99% vs 48% p<0.05” should be “9% vs 48% p <0.05”

Line 333: The authors should provide a definition for the acronym “CHW,” as it is not previously mentioned in the paper.

Line 359: The acronym ANUG, referring to the first stage of noma, is here used without explanation. Since the full term is firstly mentioned at line 85, I suggest providing a definition for the acronym there. Additionally, the acronym ANUG should be consistently used in lines 266, 359, and 437.

Lines 440-2: add reference 11

Line 445: add reference 2 for WHO definition

Line 406: I would change “quaterly” to “quaternary”

Line 494: I think “access to treatment” and “access to appropriate care” are nearly synonyms; I would rather stress the existing challenges in identifying the disease and taking action to treat it effectively. I would change line 494 to: “It is therefore not a matter of lack of access to treatment, but lack of awareness about the disease, which hinders timely diagnosis and appropriate care”

Table 3, table header: I would change “mean” to “mean/%” and “median” to “median/%”, in line with Table 2.

Table 3, 6th row: authors should clarify why only 21 participants, rather than 24, responded to the question "Can you order noma stages?"

**Summary and General Comments**

Reviewer #1: General: This is a well-described study on knowledge related to noma among health professionals in one region of Mozambique, an area not traditionally known to be endemic for this disease. The study adds to the small body of evidence around noma and health system preparedness to deal with this disease that only starts to receive more wide attention. The manuscript is well structured, easily understandable and makes clear statements. A number of comments are offered for consideration to further improve it:

- line 39 “dreadful”, line 79 “outrageous” – avoid emotional language here and throughout the manuscript

- line 59-60, 494-495: rather than “lack of access to timely and appropriate care” the issue could be called more simply “lack of diagnostic capacity” (or similar)

- line 81, line 130: “easily”, “brief” – without context it is difficult to judge what the authors claim was “easy” identification of noma cases as a short duration alone can not indicate how easy something was

- line 111-112: 70% had sub-optimal competence and half had good knowledge which adds up to over 100%

- line 119-120: the sentence should be deleted as it is offering a criticism without argument to support it

- line 126-127: this is unclear as traditional healers are said to recognize treating noma but at the same time not knowing which disease it was

- line 147: how was the study province selected?

- line 125-129: these numbers appear to be self-reported by the respondents. Or were medical records reviewed? Please clarify and discuss the option of record review in the discussion section.

- line 400-404: consider integrating this into another paragraph or deleting it as the content is not warranting a separate sub-chapter and is not relevant for the rest of the study

- line 406: “quarterly” – do the authors mean quaternary?

- Conclusions section: this should be more concise, focusing only on the key messages derived from the study

- the authors are invited to discuss in more detail the relationship between the quality of answers / correctness of diagnoses based on the vignettes and the reported awareness of the disease / previous training

- it seems that noma was part of the training curriculum of many health professionals. Do the authors have information which courses specifically? Also, it is claimed that generally, noma were not part of training (line 103) but the reported figures suggest otherwise.

- it should be discussed that while knowledge about noma, especially its early signs, was higher among well-trained staff, it is most urgently needed in peripheral facilities where affected populations seek help first

Reviewer #2: I found the paper highly engaging, well-written, and well-structured. The detailed introduction effectively explains the issues surrounding noma, making them accessible even to non-experts—a critical feature given that noma is a neglected condition. The primary limitation is the small sample size, but this and other limitations are thoroughly acknowledged within the study. I believe this study helps address the data gap on this condition, which, like in many countries, remains underdiagnosed and underreported in Mozambique. It significantly contributes to increasing awareness on noma and can serve as a starting point for promoting training and awareness-raising initiatives focused on it.

PLOS authors have the option to publish the peer review history of their article (what does this mean? ). If published, this will include your full peer review and any attached files.

**Do you want your identity to be public for this peer review?** For information about this choice, including consent withdrawal, please see our Privacy Policy .

Reviewer #1: **Yes: ** Peter Steinmann

Reviewer #2: No

---

## [Decision Letter · Decision Letter 1]

24 Feb 2025

Dear Dr. Ribes,

We are pleased to inform you that your manuscript 'Knowledge, Attitudes and Practices of Healthcare Workers Towards Noma in Zambezia, Mozambique' has been provisionally accepted for publication in PLOS Neglected Tropical Diseases.

Best regards,

Stuart Robert Ainsworth

Academic Editor

Ana LTO Nascimento

Section Editor

Shaden Kamhawi

co-Editor-in-Chief

Paul Brindley

co-Editor-in-Chief

Reviewer's Responses to Questions

**Key Review Criteria Required for Acceptance?**

**Methods**

-Are the objectives of the study clearly articulated with a clear testable hypothesis stated?

-Is the study design appropriate to address the stated objectives?

-Is the population clearly described and appropriate for the hypothesis being tested?

-Is the sample size sufficient to ensure adequate power to address the hypothesis being tested?

-Were correct statistical analysis used to support conclusions?

-Are there concerns about ethical or regulatory requirements being met?

Reviewer #1: (No Response)

Reviewer #2: Previous revisions have been thoroughly addressed. No additional comments

**Results**

-Does the analysis presented match the analysis plan?

-Are the results clearly and completely presented?

-Are the figures (Tables, Images) of sufficient quality for clarity?

Reviewer #1: (No Response)

Reviewer #2: Previous revisions have been thoroughly addressed. No additional comments

**Conclusions**

-Are the conclusions supported by the data presented?

-Are the limitations of analysis clearly described?

-Do the authors discuss how these data can be helpful to advance our understanding of the topic under study?

-Is public health relevance addressed?

Reviewer #1: (No Response)

Reviewer #2: Previous revisions have been thoroughly addressed. No additional comments

**Editorial and Data Presentation Modifications?**

Reviewer #1: (No Response)

Reviewer #2: Previous revisions have been thoroughly addressed. No additional comments

**Summary and General Comments**

Reviewer #1: The authors have adequately taken into account the points offered for consideration, and have implemented appropriate changes.

Reviewer #2: Previous revisions have been thoroughly addressed. No additional comments

PLOS authors have the option to publish the peer review history of their article (what does this mean? ). If published, this will include your full peer review and any attached files.

**Do you want your identity to be public for this peer review?** For information about this choice, including consent withdrawal, please see our Privacy Policy .

Reviewer #1: **Yes: ** Peter Steinmann

Reviewer #2: **Yes: ** Bianca Maria Longo

---

## [Editor Report · Acceptance letter]

Dear Ms Ribes,

We are delighted to inform you that your manuscript, "Knowledge, Attitudes and Practices of Healthcare Workers Towards Noma in Zambezia, Mozambique," has been formally accepted for publication in PLOS Neglected Tropical Diseases.

Best regards,

Shaden Kamhawi

co-Editor-in-Chief

Paul Brindley

co-Editor-in-Chief
